# Assessment of control strategies against *Clonorchis sinensis* infection based on a multi-group dynamic transmission model

Xiao-Hong Huang[1], Men-Bao Qian[2,3], Guang-Hu Zhu [4], Yue-Yi Fang[5], Yuan-Tao Hao[1,6], Ying-Si Lai [1,6]*

**1** Department of Medical Statistics, School of Public Health, Sun Yat-sen University, Guangzhou, Guangdong, People's Republic of China, **2** National Institute of Parasitic Diseases, Chinese Center for Disease Control and Prevention, Shanghai, People's Republic of China, **3** WHO Collaborating Centre for Tropical Diseases, Key Laboratory of Parasite and Vector Biology, Ministry of Health, Shanghai, People's Republic of China, **4** Department of Mathematics and Computing Science, Guilin University of Electronic Technology, Guilin, Guangxi, People's Republic of China, **5** Institute of Parasitic Diseases, Guangdong Provincial Center for Disease Control and Prevention, Guangzhou, Guangdong, People's Republic of China, **6** Sun Yat-sen Global Health Institute, Sun Yat-sen University, Guangzhou, Guangdong, People's Republic of China

* laiys3@mail.sysu.edu.cn

**Data Availability Statement:** All relevant data are within the manuscript and its Supporting Information files.

## Abstract

Clonorchiasis is one of the most important food-borne trematodiases affecting millions of people. Strategies were recommended by different organizations and control programmes were implemented but mostly in short-time periods. It's important to assess the long-term benefits and sustainability of possible control strategies on morbidity control of the disease. We developed a multi-group transmission model to describe the dynamics of *C. sinensis* transmission among different groups of people with different raw-fish-consumption behaviors, based on which, a full model with interventions was proposed and three common control measures (i.e., preventive chemotherapy, information, education, and communication (IEC) and environmental modification) and their possible combinations were considered. Under a typical setting of *C. sinensis* transmission, we simulated interventions according to different strategies and with a series of values of intervention parameters. We found that combinations of measures were much beneficial than those singly applied; higher coverages of measures had better effects; and strategies targeted on whole population performed better than that on at-risk population with raw-fish-consumption behaviors. The strategy recommended by the government of Guangdong Province, China shows good and sustainable effects, under which, the infection control (with human prevalence <5%) could be achieved within 7.84 years (95% CI: 5.78–12.16 years) in our study setting (with original observed prevalence 33.67%). Several sustainable strategies were provided, which could lead to infection control within 10 years. This study makes the effort to quantitatively assess the long-term effects of possible control strategies against *C. sinensis* infection under a typical transmission setting, with application of a multi-group dynamic transmission model. The proposed model is easily facilitated with other transmission settings and the simulation outputs provide useful information to support the decision-making of control strategies on clonorchiasis.

**Funding:** YSL received financial support of the Sun Yat-Sen University One Hundred Talent Grant (http://rcb.sysu.edu.cn/). This study received financial support from the CMB Open Competition Program (project no. 17-274, https://chinamedicalboard.org/), the Natural Science Foundation of Guangdong Province (project no. 2017A030313704, http://gdstc.gd.gov.cn/) and the National Natural Science Foundation of China (project no. 81703320, http://www.nsfc.gov.cn/). The funders had no role in study design, data collection and analysis, decision to publish, or preparation of the manuscript.

**Competing interests:** The authors have declared that no competing interests exist.

## Author summary

Clonorchiasis is an important food-borne parasitic disease. People get infected mainly through eating raw and infected fish, thus different behaviors of raw-fish-consumption play an important role on transmission. It's critical to find effective and sustainable control strategies for morbidity control of the disease. Control programmes have been implemented in endemic areas mostly in short periods, which is difficult to assess their long-term benefits and sustainability. We developed a multi-group model depicting transmission dynamics of the disease among different groups of people with different raw-fish-consumption behaviors, based on which, long-term effects of possible control strategies were simulated and assessed. Under a typical transmission setting, we found that combinations of control measures were much beneficial than those singly applied; higher coverages of measures had better effects; and strategies targeted on whole population performed better than that on at-risk population with raw-fish-consumption behaviors. The strategy recommended by the government of Guangdong Province, China shows good and sustainable effects. Besides, several sustainable strategies are provided under the study setting. The proposed transmission model is easily facilitated with other transmission settings. The simulation outputs can be considered together with actual practical situations to support decision making on selection of effective control strategies on clonorchiasis.

## Introduction

Clonorchiasis is one of the most important food-borne trematodiases, caused by infection with *Clonorchis sinensis* [1]. About 15 million people are infected with *C. sinensis* worldwide, predominantly located in East Asia and over 85% of infected cases happened in China [2]. Chronic infection of *C. sinensis* can lead to complications in liver and biliary systems. Furthermore, *C. sinensis* is classified as a defined carcinogenic to humans [2,3]. The global disease burden was estimated to 522,863 disability-adjusted life years (DALYs) by WHO in 2010 [4]. Therefore, control of *C. sinensis* infection becomes one of the major health challenges in endemic society.

The life cycle of *C. sinensis* involves snails as the first intermediate hosts, fish or shrimp as the second, and humans or other piscivorous mammals as the definitive hosts [5]. The most common way of human getting infected with metacercariae of *C. sinensis* is eating the raw or insufficiently cooked infected fish/shrimp, the frequency and intensity of which can directly influence the probability of getting infection [6]. Moreover, people can get infected without reporting raw-fish consumption, probably due to poor hygiene habits, e.g., eating food contaminated by metacercariae of *C. sinensis* through tableware or chopping board previously used to prepare raw fish without effective cleaning [6,7]. Different behaviors of human in raw-fish consumption may show different transmission characteristics, which is important particularly when studying the dynamics of the transmission process.

The major control measures of clonorchiasis includes preventive chemotherapy, information, education, and communication (IEC), environmental modification and the possible combinations of the above [8]. Chemotherapy is the most important measure for the management of clonorchiasis, the targeted population of which can be all residents (i.e., mass drug administration, MDA) or selective population (e.g., infected individuals and at-risk groups with raw-fish-eating behavior) [8,9]. Repeated MDA or chemotherapy with selective

population showed good morbidity control effects during short periods (e.g., 2 or 3 years) [9]. IEC combined with chemotherapy is suggested to enhance the control sustainability [10]. And environmental modification, an additional way of control measure by removing unimproved toilets built adjacent to fish ponds, can prevent the contamination of water by faeces and decrease the number of infected snails and fish [11].

Effective and sustainable control strategies on clonorchiasis are important, assisting local governments and policy makers to implement the interventions. WHO recommended a control strategy of clonorchiasis that advocates praziquantel administration for all residents every year in high endemic areas (with prevalence≥20%) and for all residents every two years or individuals regularly eating raw fish every year in moderate endemic areas (with prevalence <20%) [12]. The Chinese government set a goal in 2015 to reduce prevalence of clonorchiasis by 30% or more in areas with prevalence higher than 10% by 2020. Intervention strategy was developed, including chemotherapy every year for at-risk population with a coverage by 80%, improvement of sanitation toilets by 80% and increasing the awareness of key knowledge on clonorchiasis control among students by 95% [13]. The government of Guangdong Province, one of the most endemic provinces in China, set a more ambitious goal to reduce the prevalence by 40% in areas with prevalence higher than 20%. Stronger interventions were recommended, including coverage of chemotherapy by 80% on at-risk population every year, coverage of sanitation toilets by 90% and improvement of people's healthy behaviors by 90% [14]. However, studies were absence on evaluation of the long-term (>10 years) effects of the above strategies. Several control programmes have been implemented in endemic areas, focusing on either chemotherapy only or interventions combined with several measures [9,15–17]. But the study periods were short, mostly less than 5 years, thus it's difficult to assess the long-term effects and the sustainability. There is an urgent need to explore long-term benefits and effectiveness of different potential strategies on morbidity control of clonorchiasis within a quantitative framework.

Mathematical models play important roles for understanding the transmission dynamics of parasitic diseases [18]. Furthermore, they test hypotheses about likely dynamics and epidemiology under proposed strategies, thus support control and elimination programmes [19]. Four papers have been published on mathematical models depicting the transmission dynamics of *C. sinensis* infection. Dai and colleagues proposed a deterministic SI compartmental model describing the transmission of *C. sinensis* infection between human, snail, and fish. The authors also verified the globally asymptotically stable of the model in both disease-free and endemic equilibria [20]. Yuan and colleagues applied a deterministic SEIR and a SIR model to describe the spread of clonorchiasis in Foshan and Guangzhou, China, respectively [21,22]. Compared to Dai's model, the models developed by Yuan included recovered population as well as different status (e.g., eggs and cercariae) of the parasite. Gao and colleagues proposed a SIRS model to evaluate the effects of snail control and health education on clonorchiasis control in Foshan, China [23]. These papers analyzed relationships between model parameters and the basic production number ($R_0$), giving insights to possible directions of intervention strategies. However, no further indicators (e.g., time to achieve transmission control or extinction, percentages of prevalence decreased through time), which are more applicable for selection of specific strategies, were used for assessment. Furthermore, the models assumed homogeneity of human population, yet people with different behaviors of raw-fish consumption show different transmission patterns, which should be considered when the transmission dynamics is depicted. And control strategies focus on population with raw fish consumption instead of whole population cannot be assessed under these models. Interestingly, Bürli and colleges [24,25] developed deterministic models of *Opisthorchis viverrini* transmission dynamics to evaluate the long-term effectiveness of different interventions (e.g., MDA, IEC, and

environmental modification), by including the parameters of interventions inside the models and simulating different scenarios. *O. viverrini*, known as the other major species of liver fluke, have very similar life cycle as *C. sinensis* [26].

In this paper, with extension of Dai's transmission model and reference of Bürli's intervention model [20,24], we developed a multi-group deterministic SIS transmission model to: (1) describing the transmission dynamics among groups of population with different raw-fish-consumption behaviors and (2) quantitatively assessing the long-term effectiveness of different intervention strategies on control of *C. sinensis* infection through simulating scenarios. As a practical example, the model was applied in Fusha Town, a typical clonorchiasis endemic area with raw-fish-eating culture in Guangdong province, China. The model parameters were estimated by fitting the model with the epidemiological survey data collected in the town.

## Methods

### Method overview

We developed a basic model to describe the transmission of *C. sinensis*, with consideration of different groups of population with different raw-fish-consumption behaviors. Using the survey data in Fusha Town, Guangdong Province, we estimated the unknown model parameters under Bayesian melding approach. Validation and sensitivity analysis were undertaken afterwards. A full model with interventions were further developed by adapting the basic model with specific intervention parameters, and simulations were carried out to evaluate the long-term effects of different control strategies. In particular, we assessed: (1) the current strategies recommended by WHO, the Chinese government and the government of Guangdong Province; and (2) the three commonly used measures (i.e., preventive chemotherapy, IEC and environmental modification) when applied singly or combinedly, with different coverages, durations and frequencies of chemotherapy and improvement rates of IEC and environmental modification. The resulted sustainable intervention strategies were listed for practical application. A flow diagram of the methodology is show in Fig 1.

### The basic model

As individuals previous infected with *C. sinensis* show no apparent protective immunity after recovery from treatment [27] and re-infection is common in endemic areas [9,28], we proposed a deterministic Susceptible-Infected-Susceptible (SIS) model to describe the transmission of *C. sinensis* among human-snail-fish populations, where infected human recovered from infection will instantly return to the susceptible state. According to different frequencies of raw-fish consumption, we classified the human population into four groups, that is seldom, moderately (less than 5 times per year), often (5–10 times per year) and very often (more than 10 times per year) eating of raw fish. We assumed each group of people get infection under different transmission rates. Fig 2A describes the model structure graphically, and Tables 1 and 2 list the descriptions of state variables and model parameters, respectively. We assumed each host has two states (i.e., susceptible *S* and infectious *I*) and the transmission dynamics were modeled by a system of twelve ordinary differential equations. More detailed description of the model is shown in S1 Text.

The basic reproduction number $R_0$ (i.e., the average number of new cases generated by a typical infected individual in an otherwise susceptible population) was derived by the next

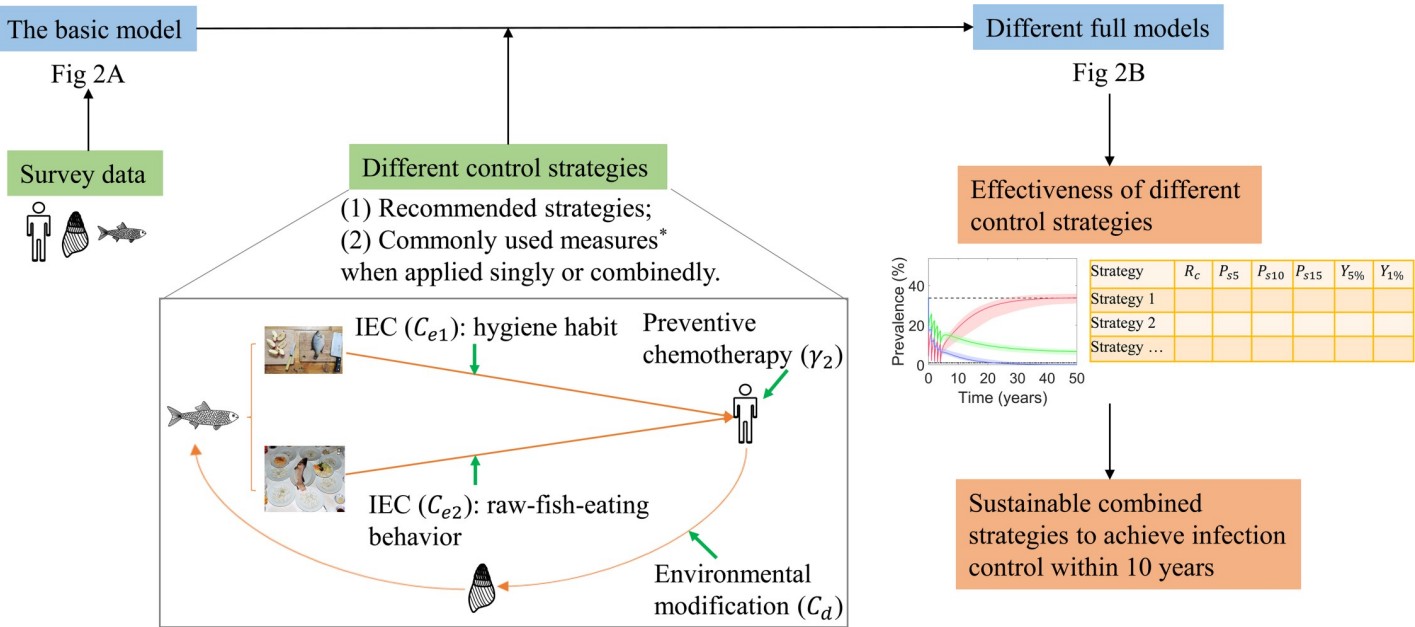

**Fig 1. The flow diagram of methodology.** [*]The commonly used interventions included preventive chemotherapy, IEC and environmental modification. Assessment is done for strategies with different coverages, durations and frequencies of chemotherapy and different improvement rates of IEC and environmental modification.

generation matrix approach (with details in S3 Text). The expression for $R_0$ is as following:

$$R_0 = \sqrt[3]{\frac{\beta_{h,1}\beta_s\beta_f\lambda_s\lambda_f(c_1\lambda_{h,1} + c_2\lambda_{h,2} + c_3\lambda_{h,3} + c_4\lambda_{h,4})}{\mu_h(\mu_h + \gamma_1)\mu_s^2\mu_f^2}}$$

## Data

As a practical example, we applied the model in Fusha Town (coordinates: 22˚40'2.64''N, 113˚20'58.2''E, area: 37 km², and population: 59,593 in the year 2017), located at the northern part of Zhongshan City, Guangdong Province (S1 Fig), where *C. sinensis* infection is highly endemic [5]. Situated among the Pearl River Delta's vast water network, the town is crisscrossed by rivers and the fishing industry is flourishing. Local residents like eating raw fish and have poor hygiene habits of using the same chopping blocks for both raw and cooked food [33]. All these geographical, environmental and socioeconomic factors make the town a typical place for the disease transmission. The prevalence data of human was obtained from a cross-sectional study done between October 2011 and June 2012 in the town, the detailed result of which is shown in Table 3 [33]. As no prevalence data of fish is available for Fusha town, we used the data from a cross-sectional survey between October 2008 to September 2009 in Zhongshan City instead, where the town belongs. The survey shows the prevalence of fish infected with cercariae of *C. sinensis* is 20.97% (39/186) [34]. Even though no survey data of snails is available for either Fusha Town or Zhongshan City, series of studies reported the prevalence of infected snail between 0.1% and 3.7% in different endemic areas of Guangdong province, with study period between 2004 to 2009 [34,35]. This interval was set as prior information for the following parameter estimation.

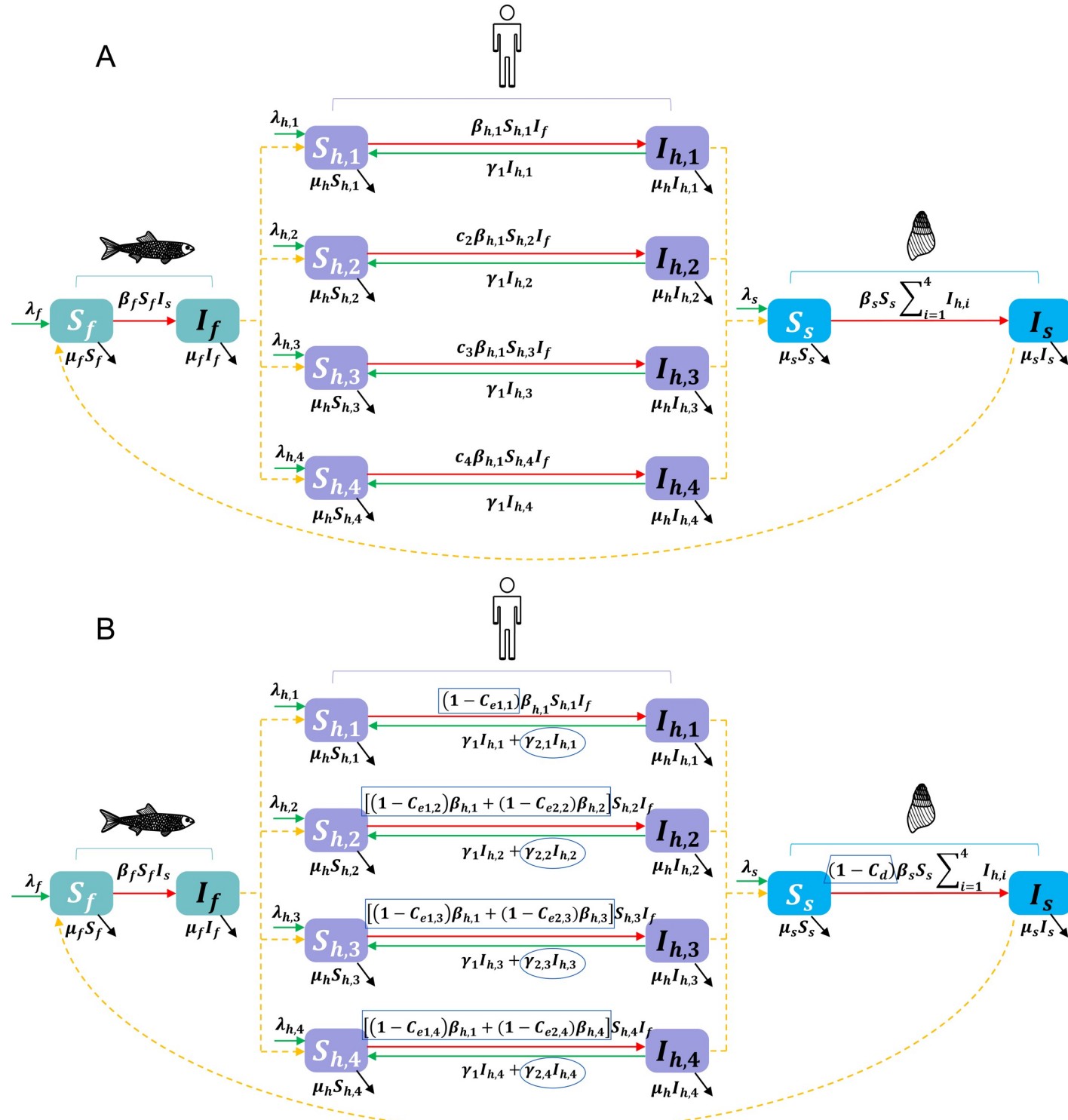

**Fig 2. The multi-group dynamic transmission models of *C. sinensis* infection.** (A) and (B) indicate the basic model and the full model with interventions, respectively. Compartments represent the population of different human groups, snails and fish; for variables and parameters see text, Tables 1 and 2. The unfilled boxes in rectangle, ellipse and trapezoidal shapes in (B) indicate interventions of IEC, chemotherapy, and environmental modification, respectively.

**Table 1. State variables used in the multi-group transmission models.**

| Variable | Description |
|---|---|
| $S_{h,1}$ | Number of susceptible humans who seldom eat raw or uncooked fish |
| $I_{h,1}$ | Number of infected humans who seldom eat raw or uncooked fish |
| $S_{h,2}$ | Number of susceptible humans who moderately eat raw or uncooked fish |
| $I_{h,2}$ | Number of infected humans who moderately eat raw or uncooked fish |
| $S_{h,3}$ | Number of susceptible humans who often eat raw or uncooked fish |
| $I_{h,3}$ | Number of infected humans who often eat raw or uncooked fish |
| $S_{h,4}$ | Number of susceptible humans who very often eat raw or uncooked fish |
| $I_{h,4}$ | Number of infected humans who very often eat raw or uncooked fish |
| $S_s$ | Number of susceptible snails |
| $I_s$ | Number of infected snails |
| $S_f$ | Number of susceptible fish |
| $I_f$ | Number of infected fish |

## Parameter estimation

In absence of time-series longitudinal data, we assumed the system of basic model on the endemic equilibrium point under the observed situation. In order to capture the local transmission dynamics, we applied Bayesian melding approach to estimate the unknown parameters based on the basic model and the survey data [36]. More details were described in the S4 Text.

Validation was undertaken by calculating the modeling efficiency statistic (EF) [37] and the coverage of observed data within 95% credible intervals of posterior estimations. Here, $\text{EF} = 1 - \frac{\sum_{i=1}^{n}(O_i-E_i)^2}{\sum_{i=1}^{n}(O_i-\bar{O})^2}$, where $O_i$ is the observed value, $\bar{O}$ is the mean of the observed values, and $E_i$ is the estimated value from the best set of estimated parameters. Chi-square test was used to see whether there is significant difference between observed and estimated prevalence [37,38]. Local sensitivity index [25] and partial rank correlation coefficients (PRCC) [39] were calculated to assess the influence of different parameters on $R_0$, locally and globally, respectively.

## Full model with interventions

We considered common types of interventions against *C. sinensis* infection, that is preventive chemotherapy, IEC, environmental modification, and possible combinations of the above measures. In particular, two kinds of improvement through IEC were considered: (1) improvement of hygiene habits and (2) changing the behavior of raw-fish-consumption. The improvements through IEC and environmental modification were assumed continuous, that is, once it happens, it will last through time in the same population. Intervention parameters were set as following: $\gamma_{2,i}$ the recovery rate of infected humans through preventive chemotherapy, $C_{m,i}$ the coverage of chemotherapy, $T_i$ the time interval of treatment, $C_{e1,i}$ the improvement rate of hygiene habits, $C_{e2,i}$ the rate of stopping raw-fish-eating behavior and $C_d$ the coverage of sanitation toilets. $i = 1,2,3,4$ represent human groups who seldom, moderately, often and very often consume raw fish, respectively. A full model is show in Fig 2B, which describes transmission dynamics of *C. sinensis* infection under different control interventions. More details are shown in S2 Text.

The approach for calculation of control reproduction number ($R_c$) in the system with interventions is similar as that of $R_0$. With the assumption that chemotherapy is distributed

**Table 2. Description of model parameters and their values (unit: day$^{-1}$).**

| Parameter | Description | Best set [95% CI] | Source |
|---|---|---|---|
| $\lambda_{h,1}$ | Recruitment of susceptible humans who seldom eat raw or uncooked fish | 0.57 [0.56–0.59] | fitting |
| $\lambda_{h,2}$ | Recruitment of susceptible humans who moderately eat raw or uncooked fish | 0.16 [0.14–0.17] | fitting |
| $\lambda_{h,3}$ | Recruitment of susceptible humans who often eat raw or uncooked fish | 0.089 [0.076–0.101] | fitting |
| $\lambda_{h,4}$ | Recruitment of susceptible humans who very often eat raw or uncooked fish | 0.045 [0.038–0.048] | fitting |
| $\lambda_s$ | Recruitment of susceptible snails | 3729.1 | [29–31] |
| $\lambda_f$ | Recruitment of susceptible fish | 1993.2 [1264.9–3606.8] | fitting |
| $\beta_{h,1}$ | Transmission rate from an infected fish to a susceptible human who seldom consumes raw fish | $2.71\times10^{-10}$ [$1.73\times10^{-10}$– $4.20\times10^{-10}$] | fitting |
| $c_2$ | Ratio of transmission rate from an infected fish to a susceptible human who moderately consumes raw or uncooked fish to that who seldom | 7.54 [6.00–10.42] | fitting |
| $c_3$ | Ratio of transmission rate from an infected fish to a susceptible human who often consumes raw or uncooked fish to that who seldom | 25.82 [18.77–28.48] | fitting |
| $c_4$ | Ratio of transmission rate from an infected fish to a susceptible human who very often consumes raw or uncooked fish to that who seldom | 333.70 [318.38–343.94] | fitting |
| $\beta_{h,2}$ | The increased transmission rate from an infected fish to a susceptible human who moderately consumes raw fish, compared to that who seldom | $1.77\times10^{-9}$ [$1.08\times10^{-9}$–$3.06\times10^{-9}$] | $\beta_{h,1}(c_2-1)$ |
| $\beta_{h,3}$ | The increased transmission rate from an infected fish to a susceptible human who often consumes raw fish, compared to that who seldom | $6.73\times10^{-9}$ [$3.51\times10^{-9}$–$9.70\times10^{-9}$] | $\beta_{h,1}(c_3-1)$ |
| $\beta_{h,4}$ | The increased transmission rate from an infected fish to a susceptible human who very often consumes raw fish, compared to that who seldom | $9.02\times10^{-8}$ [$5.76\times10^{-8}$–$1.43\times10^{-7}$] | $\beta_{h,1}(c_4-1)$ |
| $\beta_s$ | Transmission rate from an infected human to a susceptible snail | $3.31\times10^{-9}$ [$1.64\times10^{-9}$–$3.88\times10^{-9}$] | fitting |
| $\beta_f$ | Transmission rate from an infected snail to a susceptible fish | $1.53\times10^{-8}$ [$1.17\times10^{-8}$–$2.83\times10^{-8}$] | fitting |
| $\mu_h$ | Birth and death rates of human hosts | $1.49\times10^{-5}$ | [32] |
| $\mu_s$ | Birth and death rates of snails | 1/365 | [31] |
| $\mu_f$ | Birth and death rates of fish | 1/(1.58×365) [1/(2.06×365)-1/ (1.15×365)] | fitting |
| $\gamma_1$ | Basic recovery rate of infected humans through individual treatment | 0.137/365 [0.078/365-0.175/365)] | fitting |

continuously [24], $R_c$ is expressed as following:

$$R_c = \sqrt[3]{\frac{\lambda_s \lambda_f \beta_s \beta_f (1-C_d)}{\mu_s^2 \mu_f^2 \mu_h} \sum_{i=1}^{4} \frac{\lambda_{h,i}[(1-C_{e1,i})\beta_{h,1} + (1-C_{e2,i})\beta_{h,i}]}{\mu_h + \gamma_1 + \gamma_{2,i}}}$$

## Simulation of interventions

A baseline scenario was initiated at the endemic equilibrium state according to the situation of Fusha Town, which presents a typical high endemic setting. To access the current recommended strategies, we simulated the corresponding interventions recommended by WHO, the

**Table 3. Observed prevalence of *C. sinensis* infection among groups of people with different frequencies of raw fish consumption in Fusha Town [33].**

| Frequency of eating raw fish per year | No. of samples | No. of positive (%) |
|---|---|---|
| Seldom | 802 | 121 (15.08) |
| <5 times | 215 | 126 (58.60) |
| 5–10 times | 123 | 98 (79.67) |
| >10 times | 60 | 59 (98.33) |
| Total | 1200 | 404 (33.67) |

Chinese government and the government of Guangdong Province, respectively, with an intervention duration of 5 years and a further post-intervention period of 45 years. The values set for the corresponding intervention parameters are listed in Table 4. In addition, to explore the effects of intervention durations, coverage rates and frequencies of chemotherapy, improvement rates of the two kinds of healthy behaviors by IEC, improvement rates of environmental modification, and possible combinations of the three measures, we set a series of values for intervention parameters and simulated the corresponding scenarios. Indicators including control reproduction number $R_c$, prevalence and its reduced rates in 5, 10 and 15 years from the beginning of intervention, were calculated to evaluate the corresponding interventions. We considered the intervention sustainable if $R_c<1$ and the prevalence in the post-intervention period remain decreasing. In addition, with reference to the schistosomiasis control criteria in China [40], we defined areas with clonorchiasis prevalence less than 5% as infection control areas, and less than 1% as transmission control. We calculated the years from the beginning of intervention to infection control and transmission control status, respectively, for each scenario. We assumed the drug efficacy $h$ to be 92% according to the median of the corresponding studies [16,41–45].

## Results

### Parameter estimation

The estimated results of unknown parameters are shown in Table 2. The transmission rates were much higher for people who regularly eat raw fish than that for people who seldom. Particularly, the rates were 7.5 (95% CI: 6.0–10.4), 25.8 (95% CI: 18.8–28.5) and 333.7 (95% CI: 318.4–343.9) times higher for those moderately, often and very often consuming raw fish than that for those seldom, respectively. We estimated that in the study area, around 13.7% (95% CI: 7.8%-17.5%) of infected people received individual treatment and recovered per year on the endemic equilibrium point. Besides, proportions of population who seldom, moderately, often and very often consume of raw fish were estimated to 65.9% (95% CI: 64.7%-68.8%), 18.6% (95% CI: 16.3%-20.0%), 10.3% (95% CI: 8.8%-11.7%) and 5.2% (95% CI: 4.4%-5.6%), respectively. The parameter estimation shows a good fitting capacity, as it was able to correctly estimate 100% of prevalence of different hosts within 95% CI (S2 Fig), the EF score was approximately equal to 1 (*EF* = 0.9998), and the Chi-square value was 20, with *P*-value equal to 0.22 (larger than the significant level 0.05). The basic reproduction number $R_0$ was estimated to be 2.46 (95% CI: 2.31–2.56) (S3 Fig). Results for local and global sensitivity analysis of $R_0$ are show in Fig 3.

### Three recommended strategies

The three recommended strategies (i.e., strategies by WHO, the Chinese government and the government of Guangdong Province) could reduce significantly the prevalence of human by 73.56% (95% CI: 69.86%-82.56%), 57.39% (95% CI: 53.31%-62.79%) and 81.52% (95% CI: 77.16%-83.93), respectively, at the end of the 5-year interventions. The prevalence of snail and fish could be also decreased (Fig 4 and Table 5). However, the WHO recommended strategy, which only focus on chemotherapy of population, was not sustainable, as the prevalence increased after the intervention stopped (e.g., human prevalence increased by 104.15%, 95% CI: 96.68%-118.44% within 5 years of post-intervention, compared to the prevalence at the end of intervention). Strategies recommended by the Chinese government and the government of Guangdong Province, which combine chemotherapy with other measures such as environmental modification and/or IEC, performed better. Particularly, under the strategy of Guangdong Province, the study area could reach infection control and transmission control

**Table 4. Values set for intervention parameters of the current recommended strategies[*].**

| Parameter | WHO | The Chinese government | The government of Guangdong Province |
|---|---|---|---|
| $C_d$ | 0 | 0.85 | 0.90 |
| $C_{e1,1}$ | 0 | 0 | 0.90 |
| $C_{e1,2}$ | 0 | 0 | 0.90 |
| $C_{e1,3}$ | 0 | 0 | 0.90 |
| $C_{e1,4}$ | 0 | 0 | 0.90 |
| $C_{e2,2}$ | 0 | 0 | 0.90 |
| $C_{e2,3}$ | 0 | 0 | 0.90 |
| $C_{e2,4}$ | 0 | 0 | 0.90 |
| $C_{m,1}$ | 1.00 | 0 | 0 |
| $C_{m,2}$ | 1.00 | 0.80 | 0.80 |
| $C_{m,3}$ | 1.00 | 0.80 | 0.80 |
| $C_{m,4}$ | 1.00 | 0.80 | 0.80 |
| $F^{\#}$ | 1 | 1 | 1 |
| $D^{\#\#}$ | 5 | 5 | 5 |

[*]The Chinese government do have a strategy through IEC to increase the awareness of key knowledge on clonorchiasis control among students by 95%. However, such improvement does not have a direct effect on the system and is difficult to address. Thus, we set $C_{e1,i} = 0$ ($i = 1,2,3,4$) and $C_{e2,j} = 0$ ($j = 2,3,4$).

[#]The frequency of chemotherapy per year.

[##]The intervention duration of chemotherapy.

within 7.84 years (95% CI: 5.78–12.16) and 21.89 (95% CI: 17.80–33.81) years from the beginning of intervention, respectively. Hence this strategy could be more effective.

## The three common measures when applied singly

Chemotherapy, when applied singly, were able to reduce human prevalence fast during the intervention, but it was not sustainable (S3 Table). Of noted, it performed better on whole population than at-risk population with raw-fish-consumption behaviors under the same frequencies, coverage rates and duration time of intervention (S4 Fig and S3 Table). Higher frequencies of treatment, higher coverage rates and longer duration of intervention could result in larger reduction of human prevalence at the end of intervention. However, even with twice treatment each year on whole population under 100% coverage and 20 years' duration of intervention, the prevalence would still rebound after intervention stopped.

Under the assumption of continuation of the improvements, the IEC measure was sustainable when applied singly, if both kinds of improvement rates reached 93.29% (95% CI: 91.87%-94.06%) on whole population (S5 Fig and S4 Table). The effects increased when the improvement rates rose. To be noted, IEC only focus on improvement of hygiene habits of people or changing the behavior of raw-fish-consumption showed less effective than that focus on both. For example, even people achieve a 100% improvement of hygiene habits, the prevalence would remain at high level (22.08%, 95% CI: 19.50%-24.46%) after 20 years from the beginning of intervention, if no improvement was undertaken in changing people's behavior of raw-fish-consumption. Similarly, the prevalence was not able to decrease below 9.33% (95% CI: 7.07%-14.16%) within 50 years without improvement of hygiene habits, even when 100% of at-risk population with raw-fish-consumption behaviors would not consume raw fish any more. We found the effects of IEC slow, as even with a 90% improvement of both kinds of healthy behaviors, the study area needed more than 33.87 years (95% CI: 24.65–50.27 years) to reach infection control, and was not able to reach transmission control within 50 years. Similar

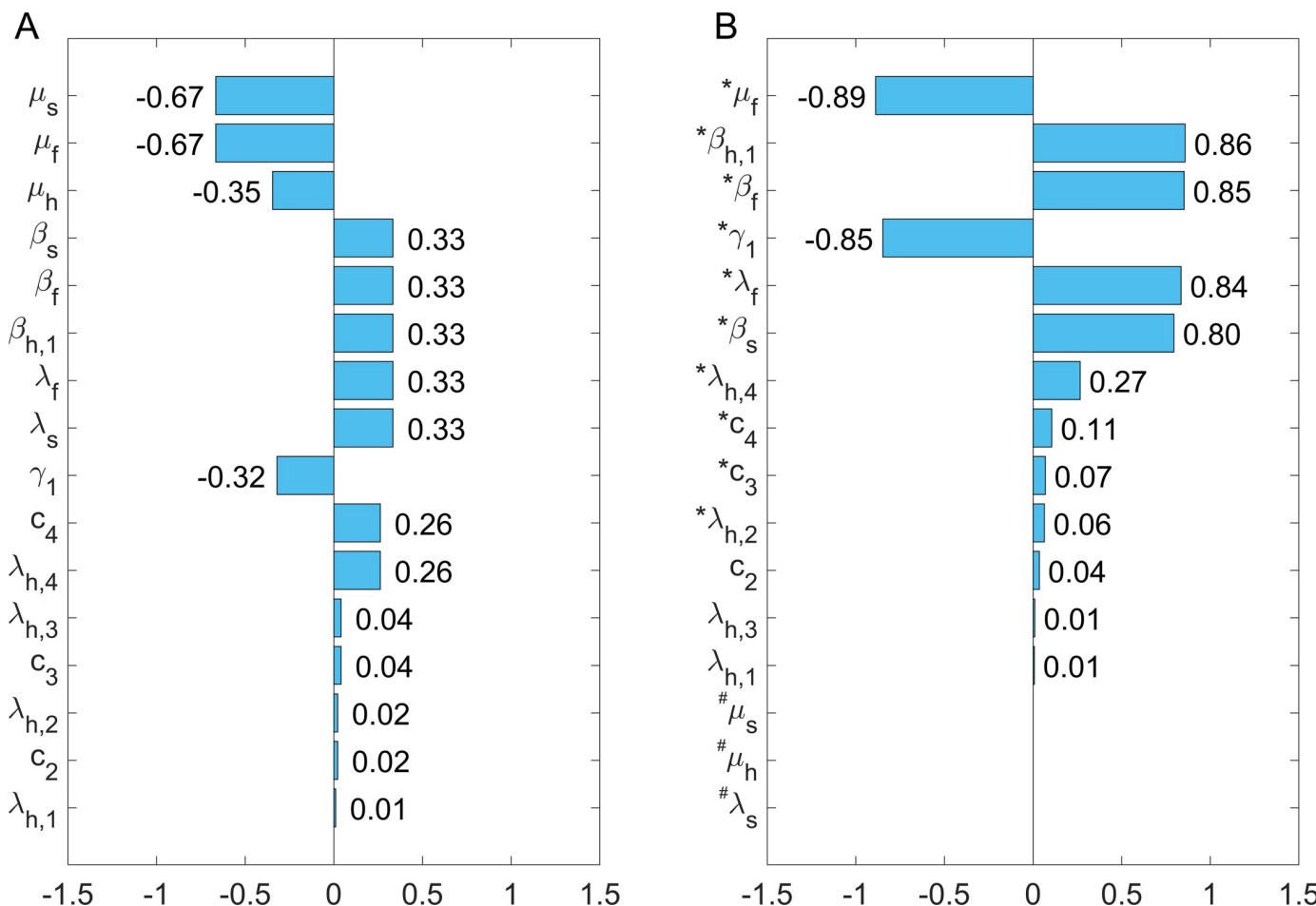

**Fig 3. Sensitivity analysis of the basic reproduction number $R_0$.** (A) Local sensitivity analysis. (B) Global sensitivity analysis. In (B), parameter with * indicates the corresponding PRCC coefficients are significantly different from zero (with *p*-values<0.05); values for parameters with # were assumed fixed in this study, thus no PRCC coefficient was calculated.

as IEC, environmental modification, when applied singly, showed slow effects on decreasing the prevalence (S5 Fig and S5 Table). For example, with a 90% coverage of sanitation toilets, the study areas needed 40.48 years (95% CI: 30.21–57.10) to reach infection control and was not able to reach transmission control within 50 years.

## Measures when applied combinedly

Generally, interventions combined with two or three measures performed much better than that only applied singly, and higher coverages resulted in larger reductions of prevalence (S6 and S7 Tables and Fig 5). Table 6 lists several selected sustainable combinations, under which the study area could achieve infection control within 10 years from the beginning of intervention. With a high coverage (i.e., 90%) of chemotherapy on whole population (once per year for five years), the improvement of people's healthy behaviors should reach 94%, or the improvement of people's healthy behaviors and the coverage of sanitation toilets should both reach 70% to achieve the goal. If chemotherapy only focus on at-risk population with raw-fish-consumption behaviors, a higher improvement (85%) of both people's healthy behaviors and coverage of sanitation toilets should be applied. Under a high coverage (90%) of sanitation toilets

 

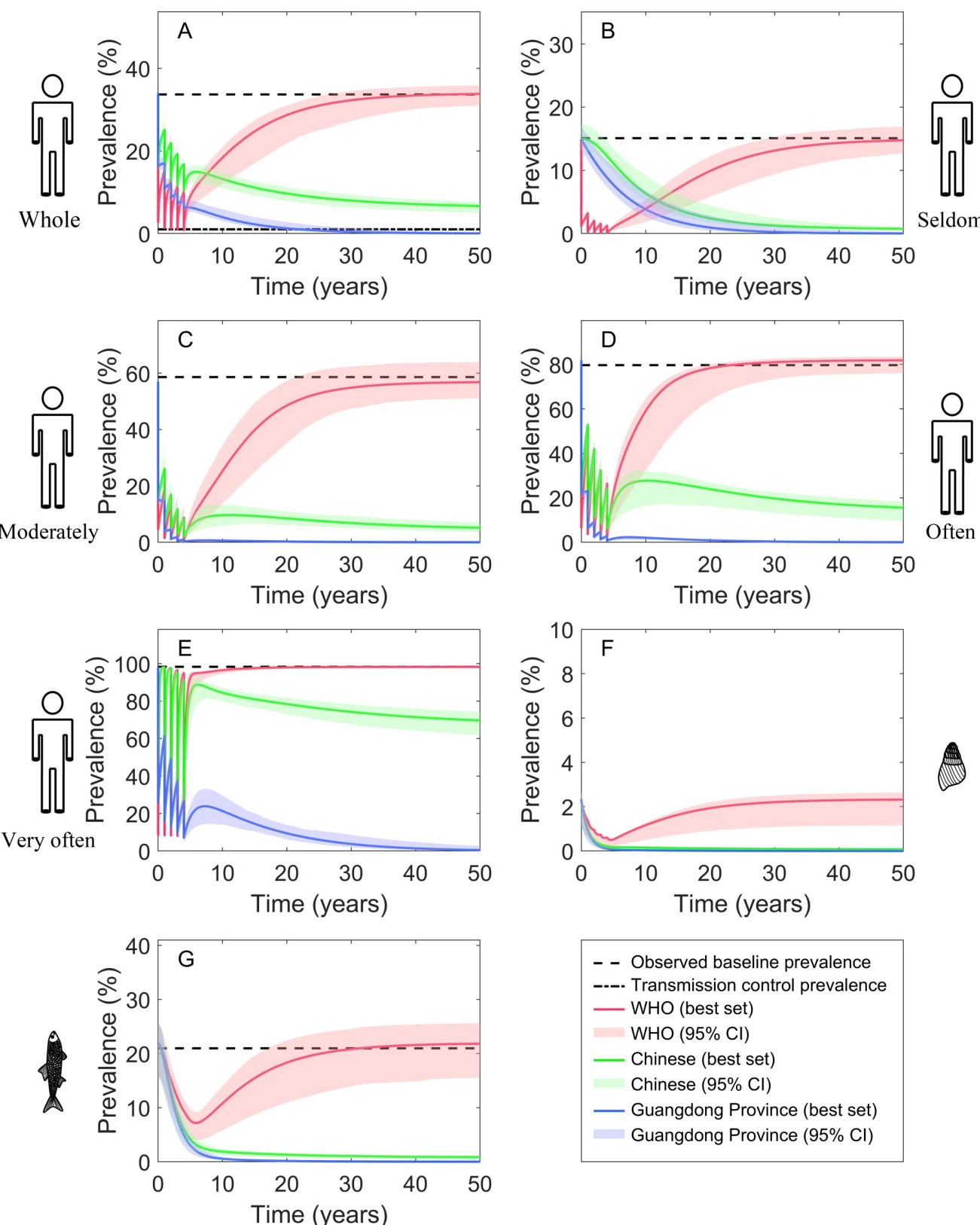

**Fig 4. Numerical simulations under the three recommended strategies.** (A) prevalence of total population, (B) prevalence of population who originally seldom eat raw fish, (C) prevalence of population who originally moderately eat raw fish, (D) prevalence of population who originally often eat raw fish, (E) prevalence of population who originally very often eat raw fish, (F) prevalence of snails, (G) prevalence of fish.

 

**Table 5. Simulation results under the scenarios of the three recommended strategies[*].**

| Indicator[#] | WHO | The Chinese government | The government of Guangdong Province |
|---|---|---|---|
| $R_c$ | 0.93 (0.77–1.00) | 0.65 (0.57–0.69) | 0.26 (0.23–0.28) |
| $P_{s5}$ (%) | 8.99 (5.68–10.33) | 14.48 (12.09–16.11) | 6.28 (5.26–8.03) |
| $P_{s10}$ (%) | 18.35 (11.98–20.93) | 13.19 (11.00–14.76) | 3.95 (3.04–5.89) |
| $P_{s15}$ (%) | 24.71 (17.04–27.35) | 11.11 (9.23–12.53) | 2.21 (1.50–3.97) |
| $r_{s5}$ (%) | 73.56 (69.86–82.56) | 57.39 (53.31–62.97) | 81.52 (77.16–83.93) |
| $r_{s10}$ (%) | 46.01 (39.56–63.70) | 61.17 (57.39–66.01) | 88.38 (82.98–90.79) |
| $r_{s15}$ (%) | 27.27 (21.38–48.33) | 67.31 (63.80–71.65) | 93.48 (88.41–95.51) |
| $Y_{5\%}$ | Never | >50 (49.71- >50) | 7.84 (5.78–12.16) |
| $Y_{1\%}$ | Never | >50 | 21.89 (17.80–33.81) |

[*]Results were expressed based on the best set of parameter estimates and their 95% CI.

[#]$P_{s5}$, $P_{s10}$ and $P_{s15}$ indicate the prevalence in 5, 10 and 15 years from the beginning of intervention, respectively; $r_{s5}$, $r_{s10}$ and $r_{s15}$ indicate the reduced rates in 5, 10 and 15 years from the beginning of intervention, compared with the baseline prevalence, respectively; $Y_{5\%}$ and $Y_{1\%}$ indicate the years to achieve infection control and transmission control from the beginning of intervention, respectively.

and high improvement (90%) of people's healthy behaviors, chemotherapy (once per year for five years) with 20% coverage on whole population or 40% on at-risk groups with raw-fish-consumption behaviors could result in transmission control within 10 years. In addition, a coverage of chemotherapy targeted on whole population by 60% combined with improvements of people's healthy behaviors and environmental modification by 70%, or a coverage of chemotherapy on at-risk population by 80% combined with improvements of people's healthy behaviors and environmental modification by 85%, could also achieve the goal.

## Discussion

### Advantages of the methodology

In this study, we developed a multi-group transmission model to describe the dynamics of *C. sinensis* transmission among people with different raw-fish-consumption behaviors. Based on the above model, we proposed a comprehensive model with interventions and simulated the long-term dynamics under various control strategies. The intervention model also allows us to investigate the control effectiveness among different raw-fish-consumption human population and with different improvements of IEC.

To our knowledge, it is the first time that a multi-group model was proposed for transmission dynamics of *C. sinensis* infection. Values of many parameter could not be obtained from existing studies. Data-driven approaches, integrating survey data into models to estimate parameters that capture the local transmission dynamics [46], are most suitable approaches of parameter estimation for our study setting. Bayesian melding approach is one of the most common data-driven approaches [36,47,48]. It has important advantages compared to the maximum likelihood approach, including the ability to produce probability distributions of the estimated parameters and model outputs which can be used for inference and projection, and the good computational efficiency when a large number of parameters are estimated simultaneously [49]. The performance of our model estimation seems good, as the outputs were able to capture the 100% of the observed data within 95% CI, the EF score was close to 1, and the Chi-square test show there was no significant difference between observed and estimated prevalence.

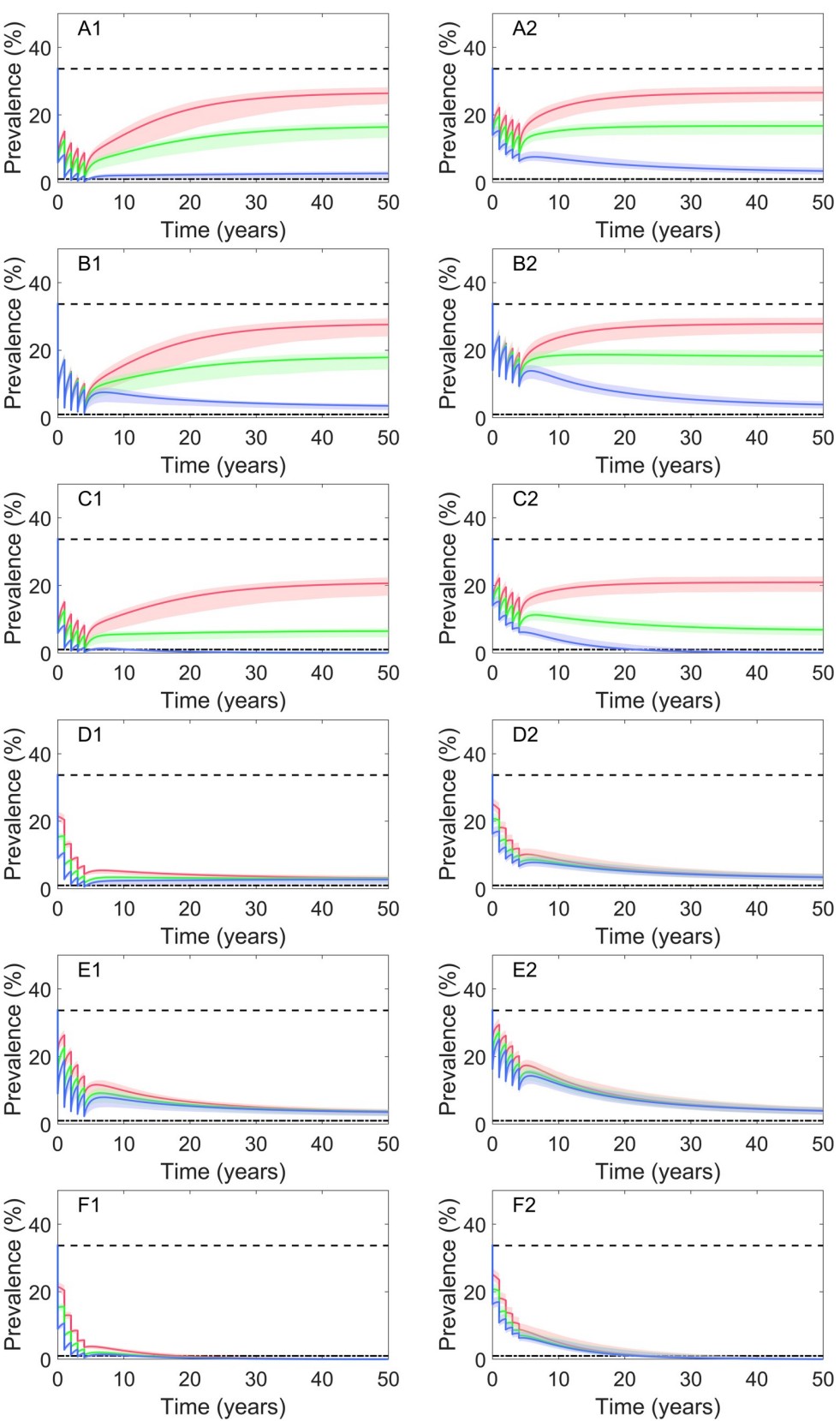

**Fig 5. Numerical simulations under different control strategies combined with two or three measures.** The left and right columns indicate chemotherapy targeted on whole population and at-risk population with raw-fish-consumption behaviors, respectively. The simulation of chemotherapy was once per year for five years. The black dotted lines indicate the observed baseline prevalence from survey data. The dash dot line indicates the transmission control prevalence (i.e., 1%). The solid lines indicate results based on best set of estimated parameters and the areas with lighter colors represent the corresponding 95% confidence intervals. (A1-A2) with coverage of chemotherapy $C_m$ = 0.90, coverage of sanitation toilets $C_d$ = 0 and improvement rate of both kinds of healthy behaviors $C_e$ = 0.30, 0.60, 0.90 (red, green, blue, respectively); (B1-B2) with $C_m$ = 0.90, $C_e$= 0 and $C_d$ = 0.30, 0.60, 0.90 (red, green, blue, respectively); (C1-C2) with $C_m$ = 0.90, $C_e$ = $C_d$ = 0.30, 0.60, 0.90 (red, green, blue, respectively); (D1-D2) with $C_e$ = 0.90, $C_d$ = 0 and $C_m$ = 0.40, 0.60, 0.80 (red, green, blue, respectively); (E1-E2) with $C_d$ = 0.90, $C_e$ = 0 and $C_m$ = 0.40, 0.60, 0.80 (red, green, blue, respectively); and (F1-F2) with $C_e$ = $C_d$ = 0.90 and $C_m$ = 0.40, 0.60, 0.80 (red, green, blue, respectively).

## Three recommended strategies

Based on a typical setting of *C. sinensis* transmission (Fusha Town), we simulated scenarios according to strategies recommended by WHO, the Chinese government and the government of Guangdong Province, with intervention duration of five years. On one hand, the WHO strategy, mainly focusing on chemotherapy of whole population, could decrease the human prevalence rapidly but were not sustainable (Fig 4), even with an extension of intervention duration to 30 years (S4 Fig), which may due to the high reinfection of the disease [9,28]. On the other hand, strategies recommended by the Chinese government and the government of Guangdong Province could reach their setting goals within 5 years of intervention (Fig 4). However, the former strategy, combined with chemotherapy and environmental modification, showed slower effects and human prevalence won't reduce to the transmission control level within 50 years. This effect might be slightly underestimated, as we ignored one component of the strategy (i.e., increasing the awareness of key knowledge on clonorchiasis control among students through IEC by 95%). Students are not among the high-risk groups of transmission [50], and it's difficult to assess how much improvement of healthy behaviors the increase of knowledge will result in. The latter strategy, even though showed the best control effects, is quite strong and ambitious, as a 90% coverage should be reached for the improvements of both environmental modification and healthy behaviors, and an 80% coverage for chemotherapy on at-risk population, which may be hard to achieve in practice, especially the one for

**Table 6. Selected sustainable combined strategies to achieve infection control within 10 years*.**

| Pop# | Strategy | | | Effectiveness | | | | | |
|---|---|---|---|---|---|---|---|---|---|
| | $C_d$ | $C_e$ | $C_m$ | $R_c$ | $P_{s5}$ | $P_{s10}$ | $P_{s15}$ | $Y_{5\%}$ | $Y_{1\%}$ |
| whole | 0.00 | 0.94 | 0.90 | 0.41 (0.34–0.44) | 0.76 (0.37–1.03) | 1.10 (0.53–1.52) | 0.98 (0.47–1.29) | 8.13 (7.46–9.17) | 13.97 (7.46–27.08) |
| | 0.70 | 0.70 | 0.90 | 0.46 (0.39–0.50) | 2.86 (1.48–3.62) | 3.82 (2.10–4.68) | 3.48 (2.03–4.13) | 7.85 (7.22–10.82) | >50 |
| | 0.90 | 0.90 | 0.20 | 0.39 (0.35–0.42) | 8.49 (7.47–9.95) | 4.96 (3.75–6.87) | 2.75 (1.82–4.72) | 9.93 (8.05–14.22) | 23.68 (19.11–36.16) |
| | 0.90 | 0.39 | 0.90 | 0.41 (0.34–0.44) | 4.44 (2.42–5.46) | 4.94 (2.94–5.99) | 3.79 (2.36–4.55) | 9.78 (6.66–13.23) | >50 (37.73- >50) |
| | 0.70 | 0.70 | 0.60 | 0.59 (0.50–0.63) | 4.54 (2.99–5.21) | 4.92 (3.35–5.66) | 4.29 (3.00–4.80) | 9.49 (6.67–13.66) | >50 |
| At-risk | 0.85 | 0.85 | 0.90 | 0.33 (0.29–0.35) | 6.79 (5.70–8.44) | 4.83 (3.80–6.63) | 3.02 (2.20–4.88) | 9.64 (7.26–14.66) | 27.33 (22.22–40.89) |
| | 0.90 | 0.90 | 0.40 | 0.34 (0.30–0.36) | 8.34 (7.23–10.22) | 4.97 (3.85–7.24) | 2.77 (1.85–4.95) | 9.96 (8.13–14.86) | 23.78 (19.36–36.71) |
| | 0.90 | 0.82 | 0.90 | 0.30 (0.27–0.32) | 7.02 (5.92–8.63) | 4.90 (3.84–6.67) | 2.95 (2.12–4.83) | 9.80 (7.46–14.53) | 25.96 (21.28–38.93) |
| | 0.85 | 0.85 | 0.80 | 0.35 (0.30–0.37) | 7.03 (5.93–8.64) | 4.95 (3.91–6.76) | 3.09 (2.24–4.96) | 9.89 (7.58–14.88) | 27.55 (22.42–41.16) |

*Results were expressed based on the best set of parameter estimates and their 95% CI; the duration of chemotherapy was 5 years; $C_d$, $C_e$ and $C_m$ indicate the coverage of sanitation toilets, the improvement rate of both kinds of healthy behaviors by IEC, and the coverage of chemotherapy on targeted population, respectively; chemotherapy is applied once per year for five years; $R_c$−$Y_{1\%}$ have the same meanings as that in Table 5.

#The targeted population of chemotherapy; "at-risk" means at-risk population with raw-fish-consumption behaviors.

improvements of people's behaviors [51]. From a practical implementation perspective, we might overestimate this strategy.

## Measures when applied singly or combinedly

In order to find more applicable strategies, we set a series of values for intervention parameters and assessed the corresponding simulated results. We found that when applied singly, IEC or environmental modification showed slow and moderate control effects but kept decreasing the prevalence of *C. sinensis* infection within 50 years, while chemotherapy indicated much quicker and stronger effects but prevalence was easy to rebound after intervention stopped. Only with very strong chemotherapy (e.g., 5 times per year for 33 years, or 2 times per year for 70 years) on whole population could lead to sustainable transmission control (S4 Fig). Besides, IEC only focus on improvement of hygiene habits of people or changing the behavior of raw-fish-consumption showed less effective than that focus on both. The higher coverages of measures resulted in better effects, which is similar as previous studies on other parasitic diseases [52]. Combinations of measures were much beneficial than those singly applied; and particularly, chemotherapy combined with both IEC and environmental modification showed much better effects than that combined with only one measure. Chemotherapy could reduce the number of infected people but could not prevent reinfections or new infections [9], while IEC helps to reduce the transmission rate from infected fish to susceptible human [25], and environmental modification decreases the chance of the transmission from infected human to snails [25]. Thus, all of the three measures implemented together will effectively control the transmission, which is strongly recommended for morbidity control of the disease. We also listed several sustainable combined strategies leading to infection control within 10 years in our current setting, with chemotherapy on either whole or at-risk population under coverages from low to high (Table 6). To be noted, durations, levels of coverage and kinds of targeted population (e.g., whole or selective) may influence the compliance of chemotherapy [9,17,53], thus both simulated effects and compliance in practice should be considered before selection of strategies.

## Insights from sensitivity analysis

Sensitivity analysis, characterizing the response of model outputs to parameter variation, shows strength and direction of the influence of different parameters on the outcomes of interest, thus provides insights on possible intervention strategies [54]. Results from sensitivity analysis (Fig 3) should be considered together with simulated outputs of transmission models and actual practical challenges, for finding better interventions. In our basic transmission model, the death rates ($\mu_h$, $\mu_s$ and $\mu_f$), the transmission rates ($\beta_{h,1}$, $\beta_s$ and $\beta_f$), the recruitments of fish and snail ($\lambda_s$ and $\lambda_f$), and the basic recovery rate $\gamma_1$ showed relatively strong influence on $R_0$. The three common measures, that is chemotherapy, IEC and environmental modification, could influence recovery rate, $\beta_{h,1}$ and $\beta_f$, respectively, which is quite reasonable to consider for interventions. Application of molluscicides may decrease $\lambda_s$ and increase $\mu_s$. However, due to the wide distribution of snails, their low prevalence of *C. sinensis* infection, and the potential toxic effects of molluscicides to fish, this measure is not recommended in endemic areas [8,55]. $\beta_f$ could be decreased by developing vaccines targeting the intermediate hosts, which is still under investigation [56]. Interestingly, parameters related to the group of people who very often consume raw fish have much stronger influence on $R_0$ compared to that of other human groups, but chemotherapy or IEC focusing only on this group showed much softer effects (S3 and S4 Tables), which may be explained by the small proportion of population in the group. On one hand, people who very often consume raw fish play an important

role in transmission and need to pay more special attention to. On the other hand, measures should be taken on wider range of people instead of just on this particular group for a better morbidity control of the disease.

## Limitations

This study is subject to some limitations. First of all, we made simplifying assumptions to several potential heterogeneities. We assumed IEC and environmental modification equally efficacious and continuous over the simulation period; however, the effects may change along with social and economic development, while it's difficult to assess how those effects might change. We ignored the impact of seasonality on the transmission, whilst temperature and rainfall, showing close relationships with seasonality, may influence the egg hatching, activities and distributions of intermediate hosts [57], thus affect the transmission of the disease. As our studied area is relatively independent from surroundings and the population are stable (S1 Table), we assumed the transmission of the disease was influenced very little by nearby areas and did not considered the interactions of disease transmission between areas. However, the distance between Fusha Town and Zhangshan City is quite close (around 29 km) and migration of population may happen, which could have an impact on the transmission. We didn't consider diversity of compliance with preventive chemotherapy on different groups of people, however, people who seldom consume raw fish may show less compliant than those very often consume. We assumed the basic transmission rate $\beta_{h,1}$ influenced mainly by improvement of hygiene habits and ignored the effect of changing raw-fish-eating behavior of people. The latter may decrease the chance of tableware or chopping board being contaminated by metacercariae of *C. sinensis* as fewer people would prepare the raw-fish-dish, thus may reduce $\beta_{h,1}$. We may underestimate the effect of this improvement. Due to unavailability of the corresponding data, we didn't address the potential heterogeneities mentioned above to avoid large uncertainties, as the performance of models with increased complexity is critically conditional on an increase of available data for parameterization [58].

Besides, we assumed human the major definitive host and ignored the contributions of reservoir hosts (e.g., dogs and cats), which may lead to slightly over-estimation of effects of the studied strategies, as they mainly focus on human. We made the parsimonious assumption that the system was at an endemic equilibrium and fitted and validated the basic model using the cross-sectional survey data, based on which parameters pertaining to the transmission process were estimated. However, we could not validate the intervention model due to lack of longitudinal data on different intervention scenarios. Preventive chemotherapy recommended by WHO is praziquantel at a dosage of 40mg/kg in one single day [45], with drug efficacy of 87% [43], while control programs in China applied different modes, commonly with praziquantel at a dosage of 75mg/kg/day for 2 days [16] or albendazole 0.8g/day for 4 days [44], the corresponding efficacies of which were around 98%-100% or 83%-92%, respectively [41,42]. Different administration modes of chemotherapy may result in different efficacies and for convenience, we used the median efficacy (92%) of the above studies. To be noted, the simulation results may have minor changes with varied drug efficacies (e.g., the simulated prevalence after five years of preventive chemotherapy under the WHO strategy would increase or decrease by around 5% if drug efficacy was set to 87% or 98%, respectively). In addition, we assessed different intervention strategies under conditions on Fusha Town, a typical setting of *C. sinensis* transmission in Guangdong Province, China, the results of which can be generalized to other endemic areas with similar epidemic and under similar socioeconomic and environmental conditions. However, for areas far away the generalization should be taken very cautiously. Nevertheless, the multi-group model can be facilitated with epidemiological data

from those areas and produce assessment results fit to their local dynamics. Besides, this study didn't consider economic factors for implementation of the interventions. We are currently extending this work to cost-effective analysis of different strategies and developing models adjusting spatial disparities to provide more comprehensive and generalized assessments.

## Conclusions

In summary, under the typical transmission setting we applied, our simulated results suggested that combinations of measures were much beneficial; higher coverages of measures had better effects; and strategies targeted on whole population performed better than that on at-risk population with raw-fish-consumption behaviors. The strategy of Guangdong Province performed best among the three recommended ones. Sustainable strategies, such as combined measures of preventive chemotherapy (once per year for five years) with 60% coverage on whole population, IEC and environmental modification with improvement rates of 70%, or combined measures of the above preventive chemotherapy with 80% coverage on at-risk population, IEC and environmental modification with improvement rates of 85%, could achieve infection control within 10 years. In conclusion, this study makes the effort to quantitatively assess the long-term effects of possible control strategies against *C. sinensis* infection. The model we proposed is facilitated to other transmission settings and the simulation outputs provide useful information that supports the selection of control strategies on clonorchiasis by researchers and decision makers.

## Supporting information

**S1 Fig. The location of Fusha Town in Pearl River Delta, Guangdong Province, China.** The red star indicates the location of the town, and the green areas indicates the Pearl River Delta. Data for boundaries of the administrative divisions were downloaded from the GADM (https://gadm.org/). The coordinates of Fusha Town and other cities was obtained through the USGS LandsatLook (https://landsatlook.usgs.gov/). Based on the above data, the maps were produced using ArcGIS 10.2.
(TIF)

**S2 Fig. Fitted and observed prevalence of different hosts.** Human1, Human2, Human3 and Human4 in X-axis indicate human groups who seldom, moderately, often and very often eat raw or uncooked fish, respectively.
(TIF)

**S3 Fig. Posterior Distribution of $R_0$ calculated by using the resampled 500 parameter vectors.**
(TIF)

**S4 Fig. Numerical simulations under applied single chemotherapy strategies with different coverages, frequencies and durations.** The parameters were set to the best set of parameter estimates. Red lines indicate chemotherapy focus on whole population while blue lines on at-risk groups with raw-fish-eating behaviors. (A) $F = 1$, $D = 5$, red line: $C_{m,1} = C_{m,2} = C_{m,3} = C_{m,4} = 0.4$, and blue line: $C_{m,1} = 0$, $C_{m,2} = C_{m,3} = C_{m,4} = 0.4$; (B) $F = 1$, $D = 5$, red line: $C_{m,1} = C_{m,2} = C_{m,3} = C_{m,4} = 0.8$, and blue line: $C_{m,1} = 0$, $C_{m,2} = C_{m,3} = C_{m,4} = 0.8$; (C) $F = 0.5$, $D = 5$, red line: $C_{m,1} = C_{m,2} = C_{m,3} = C_{m,4} = 0.8$, and blue line: $C_{m,1} = 0$, $C_{m,2} = C_{m,3} = C_{m,4} = 0.8$; (D) $F = 1$, $D = 10$, red line: $C_{m,1} = C_{m,2} = C_{m,3} = C_{m,4} = 0.8$, and blue line: $C_{m,1} = 0$, $C_{m,2} = C_{m,3} = C_{m,4} = 0.8$; (E) $F = 1$, $D = 30$, $C_{m,1} = C_{m,2} = C_{m,3} = C_{m,4} = 1.0$. (F) $F = 0.2$, $D = 33$, $C_{m,1} = C_{m,2} = C_{m,3} = C_{m,4} = 1.0$. (G) $F = 0.5$, $D = 70$, $C_{m,1} = C_{m,2} = C_{m,3} = C_{m,4} = 1.0$. $C_{m,i}$, $F$ and $D$ indicate the coverage in the $i^{th}$ group of population, treatment times per year and duration years of intervention,

respectively.
(TIF)

**S5 Fig. Numerical simulations under single strategies of IEC or environmental modification with different improvement rates.** The parameters were set to the best set of parameter estimates. The improvement rates were set to 0.2, 0.4, 0.6, 0.8, 0.9 and 1.0. (A)-(D) represent the simulations under different improvement rates of IEC: (A) IEC focus both on improvement of hygiene habits and stopping people's behavior of raw-fish-consumption; (B) IEC only focus on improvement of hygiene habits; (C) IEC only focus on stopping people's behavior of raw-fish-consumption; (D) IEC only focus on changing the behavior of raw-fish-consumption on special groups, with red line indicates people who previous ate raw fish very often changed their behavior to eat often, with green line indicates people who previous ate raw fish very often or often changed their behavior to eat moderately, and with blue line indicates people who previous have raw-fish-eating behavior stopped eat raw fish. (E) represent the simulations under different improvement rates of environmental modification.
(TIF)

**S1 Table. Resident population of Fusha Town from 2010 to 2017.**
(DOCX)

**S2 Table. Prior triangular distributions for unknown parameters and their sources (unit: $day^{-1}$).**
(DOCX)

**S3 Table. Results of simulations applied single chemotherapy strategies with different coverages, frequencies and durations targeted on different populations.**
(DOCX)

**S4 Table. Results of simulations applied single IEC with different coverages.**
(DOCX)

**S5 Table. Results of simulations applied single environmental modification with different coverages.**
(DOCX)

**S6 Table. Results of simulations applied combined strategies with different coverages and durations targeted on whole population.**
(DOCX)

**S7 Table. Results of simulations applied combined strategies with different coverages and durations targeted on at-risk population with raw-fish-consumption behaviors.**
(DOCX)

**S1 Text. The basic transmission model.**
(DOCX)

**S2 Text. The full model with interventions.**
(DOCX)

**S3 Text. The basic reproduction number $R_0$.**
(DOCX)

**S4 Text. The prior information, the likelihood function and posterior distributions of parameter estimation.**
(DOCX)

## Author Contributions

**Conceptualization:** Men-Bao Qian, Guang-Hu Zhu, Yue-Yi Fang, Yuan-Tao Hao, Ying-Si Lai.

**Data curation:** Xiao-Hong Huang.

**Formal analysis:** Xiao-Hong Huang, Ying-Si Lai.

**Funding acquisition:** Ying-Si Lai.

**Investigation:** Xiao-Hong Huang, Ying-Si Lai.

**Methodology:** Xiao-Hong Huang, Guang-Hu Zhu, Ying-Si Lai.

**Project administration:** Xiao-Hong Huang.

**Resources:** Ying-Si Lai.

**Software:** Xiao-Hong Huang, Ying-Si Lai.

**Supervision:** Ying-Si Lai.

**Validation:** Xiao-Hong Huang, Ying-Si Lai.

**Visualization:** Xiao-Hong Huang, Ying-Si Lai.

**Writing – original draft:** Xiao-Hong Huang, Ying-Si Lai.

**Writing – review & editing:** Men-Bao Qian, Guang-Hu Zhu, Yue-Yi Fang, Yuan-Tao Hao, Ying-Si Lai.

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
