## [Decision Letter · Decision Letter 0]

20 Dec 2019

Dear Dr. Lai:

Thank you very much for submitting your manuscript "Assessment of control strategies against Clonorchis sinensis infection based on a multi-group dynamic transmission model" (#PNTD-D-19-01903) for review by PLOS Neglected Tropical Diseases. Your manuscript was fully evaluated at the editorial level and by independent peer reviewers. The reviewers appreciated the attention to an important problem, but raised some substantial concerns about the manuscript as it currently stands. These issues must be addressed before we would be willing to consider a revised version of your study. We cannot, of course, promise publication at that time.

We therefore ask you to modify the manuscript according to the review recommendations before we can consider your manuscript for acceptance. Your revisions should address the specific points made by each reviewer. 

When you are ready to resubmit, please be prepared to upload the following:

(1) A letter containing a detailed list of your responses to the review comments and a description of the changes you have made in the manuscript.

(2) Two versions of the manuscript: one with either highlights or tracked changes denoting where the text has been changed (uploaded as a "Revised Article with Changes Highlighted" file); the other a clean version (uploaded as the article file).

(3) If available, a striking still image (a new image if one is available or an existing one from within your manuscript). If your manuscript is accepted for publication, this image may be featured on our website. Images should ideally be high resolution, eye-catching, single panel images; where one is available, please use 'add file' at the time of resubmission and select 'striking image' as the file type. 

Please provide a short caption, including credits, uploaded as a separate "Other" file. If your image is from someone other than yourself, please ensure that the artist has read and agreed to the terms and conditions of the Creative Commons Attribution License at http://journals.plos.org/plosntds/s/content-license (NOTE: we cannot publish copyrighted images). 

(4) If applicable, we encourage you to add a list of accession numbers/ID numbers for genes and proteins mentioned in the text (these should be listed as a paragraph at the end of the manuscript). You can supply accession numbers for any database, so long as the database is publicly accessible and stable. Examples include LocusLink and SwissProt.

(5) To enhance the reproducibility of your results, we recommend that you deposit your laboratory protocols in protocols.io, where a protocol can be assigned its own identifier (DOI) such that it can be cited independently in the future. For instructions see http://journals.plos.org/plosntds/s/submission-guidelines#loc-methods

While revising your submission, please upload your figure files to the Preflight Analysis and Conversion Engine (PACE) digital diagnostic tool, https://pacev2.apexcovantage.com/ PACE helps ensure that figures meet PLOS requirements. To use PACE, you must first register as a user. Then, login and navigate to the UPLOAD tab, where you will find detailed instructions on how to use the tool. If you encounter any issues or have any questions when using PACE, please email us at figures@plos.org.

We hope to receive your revised manuscript by Feb 18 2020 11:59PM. If you anticipate any delay in its return, we ask that you let us know the expected resubmission date by replying to this email.

To submit a revision, go to https://www.editorialmanager.com/pntd/ and log in as an Author. You will see a menu item call Submission Needing Revision. You will find your submission record there. 

Sincerely,

jong-Yil Chai

Guest Editor

Banchob Sripa

Deputy Editor

This manuscript describes about control strategies of clonorchiasis in China. It is interesting but needs some revisions before finally accepted. Please read carefully the reviewers' comments and properly address the points. Perhaps something like a flow diagram of the control measure(s) can be added for better understanding of the study. Please consider reducing unnecessary statements throughout the manuscript. The figure quality needs to be improved.

Reviewer's Responses to Questions

**Key Review Criteria Required for Acceptance?**

**Methods**

-Are the objectives of the study clearly articulated with a clear testable hypothesis stated?

-Is the study design appropriate to address the stated objectives?

-Is the population clearly described and appropriate for the hypothesis being tested?

-Is the sample size sufficient to ensure adequate power to address the hypothesis being tested?

-Were correct statistical analysis used to support conclusions?

-Are there concerns about ethical or regulatory requirements being met?

Reviewer #1: The objectives are clear, and the methods of statistical analysis seems to be good enough.

Reviewer #2: The manuscript described deterministic compartment control model for clonorchiasis by aiming for a long term outcome. Overall the objective is clear and by simulation of the model, the long term out comes can be assessed. The field data from an endemic area was used to validate the model prediction but there is no statistical support for that?.

Reviewer #3: (No Response)

**Results**

-Does the analysis presented match the analysis plan?

-Are the results clearly and completely presented?

-Are the figures (Tables, Images) of sufficient quality for clarity?

Reviewer #1: The result was described clearly.

Reviewer #2: I think for general reader who are not statistician, the results presented should be improved in a way that readers can easily follow the results. Perhaps something like a flow diagram of the control measure(s) can be added. Also for example the data in Figure 2 for human 1 ,2 and 3 in X-axis needed more clarification whether it means 3 human cases or not. I am not sure the quality of Figure 1 is adequate.

Reviewer #3: (No Response)

**Conclusions**

-Are the conclusions supported by the data presented?

-Are the limitations of analysis clearly described?

-Do the authors discuss how these data can be helpful to advance our understanding of the topic under study?

-Is public health relevance addressed?

Reviewer #1: The conclusions are definite and fine. However, it is a prediction, so the true figures will be know far later. Anyway, it is an interesting data and conclusion.

Reviewer #2: The current conclusion is wake and a more explicit conclusion is needed such that the actual outcomes from different intervention measures can be appreciated and think about utility and their effectiveness.

Reviewer #3: (No Response)

**Editorial and Data Presentation Modifications?**

Reviewer #1: No, it is fine.

Reviewer #2: (No Response)

Reviewer #3: (No Response)

**Summary and General Comments**

Reviewer #1: This is very well designed study and also well written, and a kind of innovative way of controlling clonorchiasis in China for a long term. It may be applied to other neighboring countries having clonorchiasis/ opisthorchiasis endemic area. I suppose this mathematical model is useful to implement the control measures, although in nature it may not be achieved easily. There is human food habit, which is not easily changeable by education or knowledge.

Few questions:

1) At the end of abstract: Several sustainable strategies were provided, which could lead to infection control within 10 years. -> By any strategies, clonorchiasis can be controlled down to infection control level within 10 years? Looks too simple or some additional explanations are needed.

2) Too long introduction with well know facts, and too many references are there. Hope to reduce the length and numbers of references. 

3) Maybe a map to show Fusha Town’s location is recommended. There is not significant moving in/out people from the Town. The Town seems to be close to the big city, not sure though. Is it not an important variable for this kind of analysis to predict the future?

Thank you!

Reviewer #2: The manuscript provided insightful modelling for design and choosing the right control strategy but overall the flow of the manuscript is difficult to follow for general reader. To increase clarity regarding the control measures and their effectiveness will be helpful.

Reviewer #3: It is recommended to reduce the length by deleting unnecessary sentences. 

Line 91, 96; It is not directly infected by C. sinenesis, but by metacercariae of C. sinenesis.

Lines 237-240; Even though a part of the results is based on the studies reported findings, the survey period in references should be described.

Finally, I thought it would be better to publish it in local journals.

PLOS authors have the option to publish the peer review history of their article (what does this mean?). If published, this will include your full peer review and any attached files.

Reviewer #1: No

Reviewer #2: No

Reviewer #3: No

---

## [Decision Letter · Decision Letter 1]

18 Feb 2020

Dear Dr. Lai,

We are pleased to inform you that your manuscript 'Assessment of control strategies against Clonorchis sinensis infection based on a multi-group dynamic transmission model' has been provisionally accepted for publication in PLOS Neglected Tropical Diseases.

Before your manuscript can be formally accepted you will need to complete some formatting changes, which you will receive in a follow up email. A member of our team will be in touch within two working days with a set of requests.

Best regards,

jong-Yil Chai

Guest Editor

Banchob Sripa

Deputy Editor

The revised manuscipt has been much improved according to the reviewers' comments. It seems now to be acceptable by PLoS NTD.

Reviewer's Responses to Questions

**Key Review Criteria Required for Acceptance?**

**Methods**

-Are the objectives of the study clearly articulated with a clear testable hypothesis stated?

-Is the study design appropriate to address the stated objectives?

-Is the population clearly described and appropriate for the hypothesis being tested?

-Is the sample size sufficient to ensure adequate power to address the hypothesis being tested?

-Were correct statistical analysis used to support conclusions?

-Are there concerns about ethical or regulatory requirements being met?

Reviewer #1: Methods are all adquately described.

Reviewer #2: The authors have made considerable improvement in the revised version.

**Results**

-Does the analysis presented match the analysis plan?

-Are the results clearly and completely presented?

-Are the figures (Tables, Images) of sufficient quality for clarity?

Reviewer #1: Results are all adquately described, much more clear than before.

Reviewer #2: There are additional calrification and figure.

**Conclusions**

-Are the conclusions supported by the data presented?

-Are the limitations of analysis clearly described?

-Do the authors discuss how these data can be helpful to advance our understanding of the topic under study?

-Is public health relevance addressed?

Reviewer #1: Conclusions are fine, not skewed.

Reviewer #2: OK.

**Editorial and Data Presentation Modifications?**

Reviewer #1: It is fine to accept as it is.

Reviewer #2: (No Response)

**Summary and General Comments**

Reviewer #1: The manuscipt is good although not very innovative. It is enough for publicaion at Plos NTD, I believe.

Now I would like to say only one thing. Before submiting the original paper, it would be better to check the paper thoroughly including the length of introducion, discussions or references. Revision should be minimized according to only reviewers' recommendations and comments. I suppose authors changed the whole paper more than indications given by the reviewers. I do not know the situation well, however, the authors seemed to refer the paper to someone who is more professional to prepare a revision. It is at least small violation, I suppose. Next time, hope to remind.

Thank you, and good luck!

Reviewer #2: The revised ms is still rather difficult to follow by general readers in Tropical Medicine and Parasitology.

PLOS authors have the option to publish the peer review history of their article (what does this mean?). If published, this will include your full peer review and any attached files.

Reviewer #1: Yes: Tai-Soon Yong

Reviewer #2: No

---

## [Editor Report · Acceptance letter]

19 Mar 2020

Dear Dr. Lai,

We are delighted to inform you that your manuscript, "Assessment of control strategies against Clonorchis sinensis infection based on a multi-group dynamic transmission model," has been formally accepted for publication in PLOS Neglected Tropical Diseases.

Best regards,

Serap Aksoy

Editor-in-Chief

Shaden Kamhawi

Editor-in-Chief
